# Characteristics of frequently attending children in hospital emergency departments: a systematic review

Geva Greenfield ![ORCID], Olivia Okoli, Harumi Quezada-Yamamoto, Mitch Blair ![ORCID], Sonia Saxena ![ORCID], Azeem Majeed ![ORCID], Benedict Hayhoe ![ORCID]

Primary Care and Public Health, School of Public Health, Imperial College London, London, UK

**Correspondence to**
Dr Geva Greenfield;
g.greenfield@ic.ac.uk

## ABSTRACT

**Objective** To summarise the literature on frequent attendances to hospital emergency departments (EDs) and describe sociodemographic and clinical characteristics of children who attend EDs frequently.

**Setting** Hospital EDs.

**Participants** Children <21 years, attending hospital EDs frequently.

**Primary outcome measures** Outcomes measures were defined separately in each study, and were predominantly the number of ED attendances per year.

**Results** We included 21 studies representing 6 513 627 children. Between 0.3% and 75% of all paediatric ED users were frequent users. Most studies defined four or more visits per year as a 'frequent ED' usage. Children who were frequent ED users were more likely to be less than 5 years old. In the USA, patients with public insurance were more likely to be frequent attenders. Frequent ED users more likely to be frequent users of primary care and have long-term conditions; the most common diagnoses were infections and gastroenteritis.

**Conclusions** The review included a wide range of information across various health systems, however, children who were frequent ED users have some universal characteristics in common. Policies to reduce frequent attendance might usefully focus on preschool children and supporting primary care in responding to primary care oriented conditions.

## INTRODUCTION

Demand for health and emergency services is increasing rapidly driven by a worldwide epidemiological transition with increases in life expectancy, multimorbidity and burden from long-term conditions. For example, in England, the number of hospital emergency department (ED) attendances increased by 22% between 2008/2009 and 2017/2018,[1] while the growth in National Health Service (NHS) England's funding have slowed significantly since 2010,[2] and approximately half of the NHS budget is currently spent on emergency and acute care, which is unsustainable in the long term,[3] due to growing mismatch between demand and supply. These challenges have been intensified as health services

## Strengths and limitations of this study

► This is the first systematic review we are aware of, summarising the literature on frequent attendances of children to hospital emergency departments.

► Varied definitions for what constitute a frequent attendance introduces wide variations in the proportion of frequent attenders across the studies.

► Studies originating from different countries and healthcare systems makes comparisons between them challenging.

are under increasing clinical and financial pressure amidst the COVID-19 pandemic.

Pressures on emergency health services are contributed to by a small group of individuals who attend EDs frequently, often referred to in the literature as 'Frequent attenders'.[4 5] For example, in Singapore, the total visits made by frequent users cost four times as much as for non-frequent users, representing a significant economic burden.[6] For example, in England, young children, along with the elderly, are the group with the highest number of frequent attendances.[7] Children under 5 years old, while being 5.7% of the population,[8] account for 25.1% of all frequent attenders.[7] Frequent attendance behaviour could be influenced by the emergency care models in each country, as well as the general healthcare system model and cultural differences between countries, however, there are some commodities among frequent attending patients identified in the literature.

Approximately 15% of children ED attendances have been deemed inappropriate.[9] Nevertheless, the challenge with frequent ED visits is not only the overutilisation of limited resources, but the potential harm that frequent exposure to ED can cause in terms of anxiety and distress.[10]

While frequent adult ED use has been reported in several reviews,[11 12] there are currently two reviews of frequent children's

ED attendances we are aware of. A review by Giannouchos *et al* referred to American studies, and focused on mainstream conditions, excluding specific clinical subgroups of children.[13] Poku and Hemingway focused on effectiveness of interventions for non-urgent cases.[14] This study aimed at reviewing the literature of paediatric frequent attenders in hospital EDs, focusing on the volume of frequent attendance and the demographic and clinical features of frequent attenders, with a view to better understanding how services might respond efficiently to parent and carer demand for ED services.

## METHODS

The review was performed according to the recommendation in the Cochrane Handbook for Systematic Reviews of Interventions.[15] The Preferred Reporting Items for Systematic Reviews and Meta-Analyses guideline for the reporting of systematic reviews and meta-analyses[16] was also followed.

### Eligibility criteria

Studies published in English from all over the world were included to ensure that results are generalisable. No publication year limit was placed on the search, to ensure inclusion of as many relevant studies as possible.

### Participants

Given that there is no universally accepted threshold for defining frequent ED use, we included studies which referred to children who frequently use EDs. Studies of paediatric patients aged 21 and below presenting to the ED were included. The 21 years threshold was chosen to capture as many relevant studies as possible.

### Outcome(s)

Studies had to have a measurable, definitive outcome of frequent attendances, for example, a number of attendances per child per year.

### Study design

Studies providing quantitative data on frequently attending children in ED were included. Randomized Controlled Trial (RCTs), controlled before and after studies, and interrupted time series studies were included. Cohort and cross-sectional studies related to frequent attendances of children were also included. Qualitative designs and case reports were excluded. Although observational studies are conducted in uncontrolled conditions, we included such studies in the searches as they can provide further evidence of the outcomes of frequent ED attendances of children

### Information sources

Medline, Embase, Maternity and Infant Care, PsycINFO, HMIC, Global Health and Scopus were searched using the Ovid interface.

### Search strategy

The search strategy was developed with the support of a librarian specialist. Search terms such as "emergency department", "accident adj2 emergency" were combined with search terms such as "adolescent* or child* or infant*" and "frequent admission/visit" (online supplemental appendix 1). The search strategy was adapted for each electronic database and searches were run in June 2020.

### Study selection and data collection process

The search results were transferred to Covidence (a systematic reviews management software) and further duplicates were removed. An initial screening of titles and abstracts was followed by full-text screening of papers. The screening was done independently by OO, BH and GG. Discrepancies were resolved through discussion.

### Data extraction

The extracted information was collated into a structured format (online supplemental appendix 2). Additional information including sociodemographic information, reason(s) for visiting the ED and relevant clinical information were extracted if were available.

### Assessment of risk of bias in included studies

The risk of bias of included studies was assessed using the National Heart, lung and blood institute Quality Assessment Tool for observational ccohort and cross-sectional studies and case–control studies.[17] It is an established and widely accepted Quality Assessment Tool. It was deemed appropriate because all included studies employed an observational study design. The criteria on the National Institutes of Health (NIH) Quality Assessment Tool are designed to help researchers focus on the key concepts for evaluating the internal validity of a study. They are not intended to create a list that could be tally up to arrive at a summary judgement of quality. The quality of the included studies was independently assessed by OO, BH and GG. Our responses to each of the 14 criteria are detailed in online supplemental appendix 3, followed by the criteria themselves. Discrepancies were resolved through discussion. Studies were not excluded based on risk of bias assessment.

### Data synthesis

Due to the heterogeneity in study designs and outcome measurements, a meta-analysis was deemed impractical. Instead a narrative synthesis approach was taken.[18]

### Patient and public involvement

Patients were not involved in the in the design of this study.

## RESULTS
### Study selection

Twenty-one studies representing 6513627 children met our inclusion criteria (figure 1).

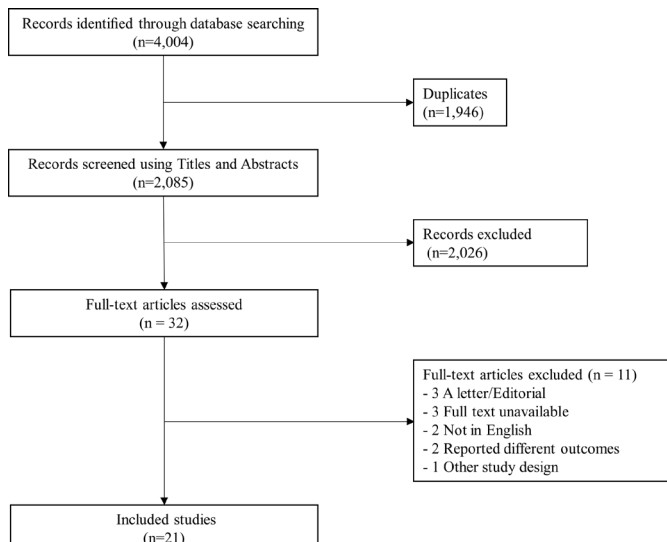

**Figure 1** PRISMA flow chart. PRISMA, Preferred Reporting Items for Systematic Reviews and Meta-Analyses.

## Study characteristics

There were 18 cohort studies[19–32] and 3 case–control studies.[33–35] Fourteen were done in the USA,[21–26 28–31 35] two in Australia,[20 33] two in Canada,[27 36] one in the UK,[32] one in Denmark[19] and one in Italy.[34] The study duration in most studies was 1 year. Frequently attending children percentage across all studies ranged from 0.3% to 75%. The data for all included studies were mainly extracted from ED registry, medical records and hospital admissions (online supplemental appendix 2).

## Risk of bias within the included studies

Overall, the quality of all the included studies was high (online supplemental appendix 3). The main limitations related to risk of bias across the studies were sample size justification or power description, measurement of exposure prior to the outcome being measured and assessment of the exposure more than once over time.

## Results of individual studies

### Definitions and rates of frequent attendances

Most studies used more than four or five ED visits in 1 year,[23–25 32 34 37] although the definition varied remarkably. The proportion of frequent attendances ranges from 0.3% to 75% of the overall number of subjects. An Australian study[20] referred to children attending extensively, more than 20 times in a year. In a review of studies from the USA, frequently attending adult and children represented approximately 4.5%–8% of all ED users and 21%–28% of all ED visits,[12] which is a narrow range than the range reported here.

### Characteristics of frequently attending children

The identified characteristics among frequently attending children were classified into three groups: demographic factors; health service factors and clinical characteristics.

### *Demographic factors*
#### Age and sex

Twelve out of 21 studies reported that children under the age of 5 were more likely to be frequent attenders compared with their older children,[20–24 26 27 31–35] Infants who were frequent attenders have a higher ED visit rate in preschool years (subsequent 2 years) than those who were not frequent attenders in infancy.[22] Five out of the 13 studies reporting gender differences found that boys were more likely than girls to attend frequently,[20 21 25 32 34] whereas four indicated that girls were more likely to attend frequently,[26 30 33 35] and one study reported no significance difference between boys and girls.[21]

#### Ethnic group

Findings regarding the association between ethnic groups and frequent attendances were mixed and specific to different cultures.[21–24 28 30–32] In six studies,[21–23 25 30 38] children from black (non-Hispanic) groups were more likely to attend EDs than children from white background. Another study reported that the proportion of white, non-Hispanic (24.7% vs 24.1%) and Hispanic (55.0% vs 53.3%) were approximately equal.[34] In another study, children from Hispanic groups was the most frequent paediatric ED users.[24] One UK study reported that children from south Asian or Asian British groups attended most frequently relatively to other groups.[32]

#### Income

Children from households with lower incomes were more likely to attend EDs frequently compared with children from households with either median or higher incomes.[27 28 30 32 39] One study[24] found an association between median household income and paediatric frequent ED usage, and another reported that actually children living in cities with an average/high income were 1.2 times more likely to have frequent non-urgent ED attendances relatively to children living in low-income cities.[34]

#### Geographical residence and proximity to an ED

Frequently attending children were more likely to live closer to the ED when compared with non-frequent ED users.[23 28 31 36 40] Four studies reported that living within an urban location[23 24 30 34] was associated with a higher likelihood of being a frequent ED user. However, another study reported that living in a rural setting was associated with frequent ED use.[28]

### *Health service and health insurance factors*
#### Health Insurance

In 9 out of the 13 studies reported on the impact of health insurance of frequent attendances, frequently attending children were using public insurance,[21 22 24–27 30–32] mainly Medicaid or State Assistance in the USA. Patients with public insurance and uninsured patients had higher rates of frequent attendance compared with patients using private or commercial insurance.[22 24 29 30 35]

### Primary care provider

Among frequent ED users, the mean number of primary care visits per year was 6.2 visits.[25] Three studies indicated that frequent users of the ED were also more likely to be frequent users of primary care compared with non-frequent ED attenders.[22–24] One study found that frequently attending children who did not have a designated primary care physician were more likely to attend the ED more frequently compared with those who did have a primary care physician.[22] This was especially the case during the first 3 years of life. Another study reported frequent ED users were more likely to live in areas with many primary care providers.[23]

### Access to healthcare barriers

Barriers to access to healthcare services, such as difficulty in finding physicians and booking appointments alongside transportation challenges, played a significant role in frequent ED use. Caregivers who experienced cultural or language barriers were 5.8 times more likely to take their child to the ED more than twice within the period of 12 months compared with children who did not have caregivers who experienced cultural or language barriers.[29] Transport challenges were also correlated with numerous ED visits.[22]

### *Clinical factors*
### Clinical diagnoses

Having chronic conditions were associated with high frequency ED use.[21 23 25 26 28 29 31–33] The most common diagnoses among frequently attending children were upper respiratory tract infections,[20 21 23 25–27 34] accounting for approximately one-third of all frequent attendances. Other common diagnoses were viral infections,[20 23 27] gastroenteritis[21 23 25] and mental health problems.[36] Frequently attending children were also more likely to have a discharge diagnosis related to oncology, neurology, respiratory, endocrinology and psychiatric complaints, compared with occasional attenders who were more likely to present with injury or trauma.[21 33] However, a Hawaiian study recognised that patients with chronic conditions often presented to the ED with problems which were not related to their chronic condition.[31] An American study identified various conditions associated with frequent ED attendances in children, such as a history of increased tone (representative of neuromuscular disease), eczema, pneumonia, vocal cord dysfunction, technology dependence (gastrostomy tube or tracheostomy) and various allergies. Frequently attending children in this study had a higher likelihood of acquiring prescriptions for asthma medication or medication related to other diseases.[30]

### Physical injuries and maltreatment

Frequently attending children presented with superficial injuries such as dislocations, sprains, strain injuries, contusion, open wounds and other trauma to the skin.[19 23] Most of these incidents occurred in children aged 6–11 years old and adolescents.[34] Young children with higher frequency of ED visits are at significantly increased risk of being victims of child maltreatment compared with occasional users.[35]

### Arrival, admission and discharge

Frequently attending children were more likely to arrive to the ED by ambulance compared with occasional attenders.[20 33] They were more likely to have at least more than one ED visit leading to hospitalisation and/or transfer to another department, compared with non-frequent ED utilisers.[21 24 33] They were two times more likely to require admission for a mental health-related problem, compared with non-frequent ED utilisers.[33] They were 1.2 times more likely more likely to have received antibiotics and laboratory testing on discharge, compared with infrequent ED utilisers.[24]

## DISCUSSION
### Summary of main findings

The percentage of children who were frequent ED users ranged from 0.3% to 75%. Children who were frequent ED users were more likely to be aged less than 5 years, live in an urban location, near to an ED. Findings regarding sex, income and ethnicity were mixed. In the USA, patients with public insurance were more likely frequent attendance compared with patients using private insurance. Frequent ED users more likely to be frequent users of primary care compared with non-frequent ED attenders. They tended to have long-term conditions and the most common diagnoses among frequently attending children were upper respiratory tract infections, viral infections and gastroenteritis.

### Strengths and limitations

To our knowledge, this is the first systematic review to report on frequent ED attendance of children in a global perspective. Some challenges in the findings should be acknowledged. All the studies had observational design. The varied definitions for what constitute a frequent attendance, challenge the comparability of the studies, introducing wide variations in the proportion of frequent attenders between the studies. It is difficult to know whether the source of this variation is an outcome of differences in measurement methods, differences in populations, differences in healthcare systems structure or other sources. Likewise, the studies originated from different countries, cultures and healthcare systems, which makes comparisons between them challenging. Whereas most studies included all children <18 years, some studies focused on specific groups such as the <5 years old. Also, there was no blinding of the outcome assessor neither did majority of studies report whether there was a loss to follow-up. Finally, this systematic review restricted literature to only those that reported in English, which could potentially introduce selection bias.

## Implications for future research

There is still little evidence on whether frequent ED users attend multiple ED sites or whether they are frequent users of other healthcare services. Linked data sets containing primary, secondary, hospital and social care data could be used for this purpose.

It would be useful to look at parent/carer characteristics as factors affecting frequently attending children. This is particularly important with younger children where the decision to attend the ED is made by the carer. This might highlight specific parent groups suspectable to frequent ED use.

From methodological perspective, using a uniform definition which will enable comparison across different studies. An alternative approach to the commonly used visit count approach using a proportional threshold[41] (eg, top 10% of all enlisted frequently attending children).

No data were found on any association between frequent ED attendances of children and risks of infection and other adverse health events following frequent ED visits. This is an important perspective to be looked at in future studies. Likewise, the fact that the studies come from high income countries might reflect a tendency for the frequent attendances behaviour to be more prominent matter in high-income countries, however, it might also reflect little research interest in frequent attendance behaviour in Low and Middle-income Countries (LMIC). This might call for attention for frequent attendance in LMIC if challenges exist.

In relation to ethnicity, the findings were mixed. Ethnicity itself might manifest cultural differences and different attitudes toward the healthcare system and to medicine in general. While is it challenging to attribute clear characteristics to each ethnicity, the ethnicity still played a role in predicting frequent attendances. A more useful indicator was the association of frequent attendances with lower household income. A lower household income may imply stronger reliance on public insurance and public healthcare services, as was evident in the findings. Public healthcare services enable accessible healthcare to all or most people, however, the lack of cost containment may contribute to frequent attendances. It is crucial though to establish the reasons for frequent attendances, whether they manifest from unresolved clinical matters, lack of self-care skills, mental health challenges or broader social determinants.

## Implications for healthcare policy and practice

Creating tailored pathways to provide a response to the specific needs of frequently attending children is crucial. The characteristics of frequently attending children highlighted by this study can be used by healthcare providers to develop targeted interventions aimed at providing tailored response to the needs of frequent ED users either through primary care or other healthcare services. EDs could highlight paediatric patients attending EDs frequently to their general practitioner (GP) practice so their GP could identify and seek to address unmet needs.

Proactive identification of children who frequently attend EDs at the primary care end (through flagging ED discharge summaries, or through EDs highlighting frequent attendance to the GP), and reviewing their situation to identify possible areas of concern or failure in support, could potentially reduce reliance on ED. Some ED triage services in England already take this approach, where a triage nurse may admit the patient to the ED or may redirect the patient to their GP, to an urgent care centre, a well-being service or social care service.

Frequent attendances might be a feature of publicly funded systems where services are widely available to patients at no charge. Indeed, US data show that patients with public insurance and the uninsured had higher rates of frequent attendance compared with those using private or commercial insurance.

But rather than trying to contain frequent attenders using caps and charges, it might be useful to understand why frequent users keep coming to the ED, and whether frequent attendances stem from pure clinical matters, psychosocial factors, lack of health literacy or self-care skills, for example, among first time parents, or other factors.

Access to primary care is cited as a main driver in ED visits in children, particularly during out of hours.[3 42] While access to primary care cannot be directly implied as a reason for frequent attenders, it might be a potential explanation for frequent attendances. If parents face challenges in accessing primary care, particularly during out-of-hours, they will understandably seek hospital care. Ensuring that primary care is accessible and effective in meeting the needs of children and their carers will, therefore, be essential in reducing unnecessary paediatric ED use. Children who use primary care benefit from health service accessibility, health prevention and early management of illnesses.[43] Improving access to primary care could hence reduce repeat ED use and improve overall long-term health.

Furthermore, as some clinical characteristics have been identified among frequently attending children, policymakers can implement national campaigns that will create awareness of what healthcare provider to visit with the appropriate associated symptoms.

## CONCLUSION

The review included a wide range of information across various health systems, however, children who were frequent ED users have some universal characteristics in common. We identified demographic factors, health service factors and clinical characteristics relating to frequent ED attendances among children.

Policies to reduce frequent attendance might usefully focus on preschool children and supporting primary care in responding to primary care oriented conditions. Further investigation of the potential of tailored pathways to respond to the specific health needs of frequently attending children will be valuable in identification of

effective interventions to reduce frequent attendance and improve children's long-term health.

**Contributors** GG, OO and BH were involved with conception and design, conducted the data analysis, and drafted the manuscript. HQ-Y was involved in conducting the literature searches, screening, data extraction and synthesis, and revised various versions of the manuscript. MB, SS and AM were involved with acquisition of funding, conception and design, interpretation of the findings, provided clinical perspective and revised various versions of the manuscript. GG is the guarantor of this study.

**Funding** This report is independent research supported by the National Institute for Health Research Applied Research Collaboration Northwest London. Grant No. NIHR200180. HQ-Y was supported by the WHO Collaborating Centre for Public Health Education and Training at Imperial College London.

**Disclaimer** The views expressed in this publication are those of the authors and not necessarily those of the National Institute for Health Research or the Department of Health and Social Care.

**Competing interests** None declared.

**Patient consent for publication** Not applicable.

**Provenance and peer review** Not commissioned; externally peer reviewed.

**Data availability statement** All data relevant to the study are included in the article or uploaded as online supplemental information.

**ORCID iDs**
Geva Greenfield http://orcid.org/0000-0001-9779-2486
Mitch Blair http://orcid.org/0000-0001-7442-0188
Sonia Saxena http://orcid.org/0000-0003-3787-2083
Azeem Majeed http://orcid.org/0000-0002-2357-9858
Benedict Hayhoe http://orcid.org/0000-0002-2645-6191

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
