## [Reviewer comments · BMJ Open]

ARTICLE DETAILS

TITLE (PROVISIONAL)	Characteristics of frequently attending children in hospital emergency departments: A systematic review
AUTHORS	Greenfield, Geva; Okoli, Olivia; Quezada-Yamamoto, Harumi; Blair, Mitch; Saxena, Sonia; Majeed, Azeem; Hayhoe, Benedict

VERSION 1 – REVIEW

REVIEWER	Doğan, Nurettin Kocaeli Universitesi
REVIEW RETURNED	29-May-2021

GENERAL COMMENTS	I evaluated the article entitled "Characteristics of frequently attending children in hospital emergency departments: A systematic review", which was sent to the BMJ Open for publication. This article is a well-written systematic review of a current topic. I thank the authors for not trying to perform a meta-analysis on such a topic. My comments about the article are as follows: 1. Introduction: Well written overall. I would expect such a review to emphasize the impact of cross-cultural differences on emergency department readmission behavior in the Introduction.2. Methods: The subheadings of the "Methods" did not seem appropriate for a systematic review. The indexes written in the "Information Sources" section should have been moved in the "eligibility criteria" section. It should also be stated which interfaces are used for scanning the databases.3. Results: In the first part of the Results, it can be stated the origin of the articles (from which index?), if possible, in the flowchart.4. Results: In evaluating the results of studies with a low risk of bias, a figure in the form of a map weighted by the number of patients can be presented.5. Discussion: Well written. Kind regards.
--

REVIEWER	Chang, Nien-Tzu National Taiwan University
REVIEW RETURNED	11-Jun-2021

GENERAL COMMENTS	Review of BMJ open 051409 manuscript
--------------------------------------

	Thank you for the opportunity to review the manuscript entitled " Characteristics of frequently attending children in hospital emergency departments: A systematic review". The authors aimed to summarize the ED utilization and users' characteristics using systematic review method. Overall, please edit the methods, results, and discussion to better portray the value of your study as well as describe your vision of future research on the topic.  - Introduction: was good written - Methods: Meta analysis and subgroup analysis should be added from 21 studies, 6 million children. Authors claim to include RCT studies into the collection, but the results have no shown. - Results: The order of the included papers were not clear, it should be illustrated by a flowchart to describe how many papers included or duplicated between all data sources (Medline, Embase, Maternity and Infant Care, psycINFO?...etc.) Please follow the PRISMA guidelines for the full SR description. - Discussion: the current analysis and presented data set forbid any kind of meaningful conclusion. - The error (order of the paper lists) in the Appendix 2. is needed to be corrected. I reviewed the article carefully. This study is a very simple review. All statements and findings are well known. There is no new contribution to the literature for clinical impact. In conclusion, the manuscript quality is not good enough for publication. It should be rejected at this stage.
--	--

REVIEWER	Johnson, Leigh Bristol Medical School, Population Health Sciences
REVIEW RETURNED	27-Jun-2021

GENERAL COMMENTS	A high proportion of emergency care episodes are for children. Describing characteristics that may help health providers develop effective interventions is important so this review is interesting and welcome. The review features studies presented in English from Europe, North America and Australia so is not quite global as it is described on page 14 line 8. There are several typing/grammatical errors – I'll list these later and make general comments first. p.5 line 11 – perhaps this first sentence needs to be referenced? p.5 line 20 – “unsustainable in the long term”. Perhaps you could identify the reason this is unsustainable. Is it because of a shortfall of primary care funding as reference 3 suggests? p.5 line 32 – the sentence “In some health systems...” is referenced by a single study that is specific to Singapore.
---

	p. line 49 – Reference 9 concludes that there is no increased risk of infection from ED visits which contradicts rather than supports the text. Reference 10 is a cohort study of elderly patients – I wonder if this applicable to this review? p.11 line 8-12 City of residence cannot be presented as a descriptor of individual level socioeconomic status (SES). A very wide range of individual SES will be evident in any city. The original study may have presented this but perhaps this should not be included in the review as a descriptor of SES. p.12 “PCP” – perhaps this should be defined. p. 12 “Allergies were less common” – this is unclear, less common in what sense, or compared to which other group? p.13 line 45 Socioeconomic status is a key patient characteristic that merits full discussion. The summary results of each study are not fully visible (the page is not wide enough to include them), but there seems to be numerous mentions of SES and also ethnicity. These key characteristics should be discussed as fully as possible but are almost missing from the discussion. Typing and grammar for correction: p.5 line 16 “funding have slowed significantly” p.9 line 36 “Measurement” – capital M. p.10 line 43 “study reported proportion” p.11 line 3 “Residence in a poverty” p.11 line 13 “studies reported on health insurance” p.12 line 44 “A Hawaiian study” – capital A. p.13 line 18 “more likely to arrive the ED” p.14 line 56 “definition of will” – missing word.
--	--

VERSION 1 – AUTHOR RESPONSE

Reviewer: 1

Dr. Nurettin Doğan, Kocaeli Universitesi

1. Introduction: Well written overall. I would expect such a review to emphasize the impact of cross-cultural differences on emergency department readmission behavior in the Introduction.

Thank you for this comment. We added a sentence at the end of the first paragraph of the introduction:

“Frequent attendance behaviour could be influenced by the emergency care models in each country, as well as the general healthcare system model and cultural differences between countries, however there are some commodities among frequent attending patients identified in the literature.”

2. Methods: The subheadings of the "Methods" did not seem appropriate for a systematic review. The indexes written in the "Information Sources" section should have been moved in the "eligibility criteria" section. It should also be stated which interfaces are used for scanning the databases.

The subheadings of the "Methods" was used as instructed by the PRISMA checklist for systematic reviews. Likewise, the title "Information Sources" is instructed by the guidelines as an expected subheading under the methods section. We added details about the inface used for searching.

3. Results: In the first part of the Results, it can be stated the origin of the articles (from which index?), if possible, in the flowchart.

This could be done however we wonder what value it will provide to the reader. PRISMA guidelines do not require such level of reporting, for the reason that the value of a systematic review is in the aggregation of studies accumulated across multiple databases rather than which database were they found in.

4. Results: In evaluating the results of studies with a low risk of bias, a figure in the form of a map weighted by the number of patients can be presented.

We are not sure what is the purpose of such map and how weighting by the number of patients would provide any useful insight. The low risk of bias can be influenced by many other factors other than the number of patients who participated in the study. This chart would have been more relevant if a meta-analysis was relevant to the study.

5. Discussion: Well written.

Reviewer: 2

Dr. Nien-Tzu Chang, National Taiwan University

- Introduction: was good written

- Methods: Meta analysis and subgroup analysis should be added from 21 studies, 6 million children.

[Editor: it's not clear what the reviewer means as you have stated that meta-analysis was not appropriate and the number of studies/children is given in the results (correctly)]

It is unclear how will a meta-analysis contribute to answering the study question. The number of participants across all studies is not too relevant if studies do not report similar outcomes which could be used for a meta-analysis.

Authors claim to include RCT studies into the collection, but the results have no shown.

We include RCTs as a study design in the eligibility criteria however this does not necessities such studies will be found in the actual searches.

- Results: The order of the included papers were not clear, it should be illustrated by a flowchart to describe how many papers included or duplicated between all data sources (Medline, Embase, Maternity and Infant Care, psycINFO?...etc.)

Please refer to Figure 1. Per PRISMA guidelines the total number of studies searched and deduplicated is reported. The value of a systematic review is in the aggregation of studies accumulated across multiple databases rather than which database were they found in.

Please follow the PRISMA guidelines for the full SR description.

Please refer to the PRISMA Statement attached to the submission. We would appreciate if the reviewer could provide more specific input if any specific elements were not described properly.

- Discussion: the current analysis and presented data set forbid any kind of meaningful conclusion.

In the absence of a systematic review on the topic we are confident the review provides meaningful summary of the current evidence on the topic of paediatric frequent attendances in EDs. We are yet mindful that there might be elements which could be improved in the discussion and would be grateful if the reviewer could point out to specific elements in the analysis and data which forbid meaningful conclusions.

- The error (order of the paper lists) in the Appendix 2. is needed to be corrected.

We would be grateful if the reviewer could point out what is the error in the table. The list is sorted out according to year of publication.

I reviewed the article carefully. This study is a very simple review. All statements and findings are well known. There is no new contribution to the literature for clinical impact. In conclusion, the manuscript quality is not good enough for publication. It should be rejected at this stage.

[Editor: clearly we do not agree with this statement but please emphasise what this study adds to the literature].

As mentioned above, in the absence of a systematic review on the topic we are confident the review provides meaningful summary of the current evidence on the topic of paediatric frequent attendances in EDs, and would appreciate more specific input about elements of this study which could be improved.

Reviewer: 3
Mr. Leigh Johnson, Bristol Medical School

Thank you for your review and helpful suggestions. The search strategy did not exclude any studies based on geographic location, so the included studies come from any country where a relevant study was performed. Indeed, the studies emerged from the USA, Australia, Canada, UK, Denmark, Italy, with not representation for LMIC, however this what the data tell us. We have now referred to it in the "Implications for future research" in the discussion:

"The fact that the studies come from high income countries might reflect a tendency for the frequent attendances behaviour to be more prominent matter in high-income countries, however it might also reflect little research interest in frequent attendance behaviour in LMIC."

There are several typing/grammatical errors – I'll list these later and make general comments first.

p.5 line 11 – perhaps this first sentence needs to be referenced?

We have added 2 references to support the sentence.

p.5 line 20 – “unsustainable in the long term”. Perhaps you could identify the reason this is unsustainable. Is it because of a shortfall of primary care funding as reference 3 suggests?

We have added the growing mismatch between demand for healthcare services and supply of funding as a reason for unsustainability.

p.5 line 32 – the sentence “In some health systems...” is referenced by a single study that is specific to Singapore.

We have corrected the sentence to refer specifically to Singapore rather than have a general statement.

p. line 49 – Reference 9 concludes that there is no increased risk of infection from ED visits which contradicts rather than supports the text. Reference 10 is a cohort study of elderly patients – I wonder if this applicable to this review?

We agree this is a controversial matter and have removed the sentence as it feels the main point has been made.

p.11 line 8-12 City of residence cannot be presented as a descriptor of individual level socioeconomic status (SES). A very wide range of individual SES will be evident in any city. The original study may have presented this but perhaps this should not be included in the review as a descriptor of SES.

The study (Riva et al.) referred to the average income range of the city of residence. We merely report their findings.

p.12 “PCP” – perhaps this should be defined.

We have revised the sentence to: “...frequent ED users were more likely to live in areas with many primary care providers.”

p. 12 “Allergies were less common” – this is unclear, less common in what sense, or compared to which other group?

The sentence has been revised accordingly:

“An American study identified various conditions associated with frequent ED attendances in children, such as a history of increased tone (representative of neuromuscular disease), eczema, pneumonia, vocal cord dysfunction, technology dependence (gastrostomy tube or tracheostomy), and various allergies. Frequently attending children in this study had a higher likelihood of acquiring prescriptions for asthma medication or medication related to other diseases [30].”

p.13 line 45 Socioeconomic status is a key patient characteristic that merits full discussion. The summary results of each study are not fully visible (the page is not wide enough to include them), but there seems to be numerous mentions of SES and also ethnicity. These key characteristics should be discussed as fully as possible but are almost missing from the discussion.

The table has been reformatted to fit a page. We have added the following text to refer to the income and ethnicity with the word limit.

“In relation to ethnicity, the findings were mixed. Ethnicity itself might manifest cultural differences and different attitudes toward the healthcare system and to medicine in general. While it is challenging to attribute clear characteristics to each ethnicity, the ethnicity still played a role in predicting frequent attendances. A more useful indicator was the association of frequent attendances with lower household income. A lower household income may imply stronger reliance on public insurance and public healthcare services, as was evident in the findings. Public healthcare services enable accessible healthcare to all or most people, however the lack of cost containment may contribute to frequent attendances. It is crucial though to establish the reasons for frequent attendances, whether they manifest from unresolved clinical matters, lack of self-care skills, mental health challenges or broader social determinants. “

Typing and grammar for correction:

p.5 line 16 “funding have slowed significantly”

p.9 line 36 “Measurement” – capital M.

p.10 line 43 “study reported proportion”

p.11 line 3 “Residence in a poverty”

p.11 line 13 “studies reported on health insurance”

p.12 line 44 “A Hawaiian study” – capital A.

p.13 line 18 “more likely to arrive the ED”

p.14 line 56 “definition of will” – missing word.

All of these have been corrected.